# Analysis of Circulating Tumour Cells in Early-Stage Uveal Melanoma: Evaluation of Tumour Marker Expression to Increase Capture

**DOI:** 10.3390/cancers13235990

**Published:** 2021-11-28

**Authors:** Aaron B. Beasley, Timothy W. Isaacs, Tersia Vermeulen, James Freeman, Jean-Louis DeSousa, Riyaz Bhikoo, Doireann Hennessy, Anna Reid, Fred K. Chen, Jacqueline Bentel, Daniel McKay, R. Max Conway, Michelle R. Pereira, Bob Mirzai, Leslie Calapre, Wendy N. Erber, Melanie R. Ziman, Elin S. Gray

**Affiliations:** 1School of Medical and Health Sciences, Edith Cowan University, Joondalup, WA 6027, Australia; a.beasley@ecu.edu.au (A.B.B.); jamie_nocturne@hotmail.com (J.F.); anna.reid@ecu.edu.au (A.R.); michelle.r.pereira923@gmail.com (M.R.P.); l.calapre@ecu.edu.au (L.C.); m.ziman@ecu.edu.au (M.R.Z.); 2Centre for Precision Health, Edith Cowan University, Joondalup, WA 6027, Australia; 3Perth Retina, Subiaco, WA 6008, Australia; tim@perthretina.com; 4Centre for Ophthalmology and Visual Science (Incorporating Lions Eye Institute), The University of Western Australia, Perth, WA 6000, Australia; jlds@lei.org.au (J.-L.D.); riyazbhikoo@gmail.com (R.B.); fredchen@lei.org.au (F.K.C.); 5Department of Ophthalmology, Royal Perth Hospital, Perth, WA 6000, Australia; doireann.hennessy@sjog.org.au; 6Anatomical Pathology, PathWest Laboratory Medicine, Fiona Stanley Hospital, Murdoch, WA 6150, Australia; tersiav1@optusnet.com.au (T.V.); jacqueline.bentel@health.wa.gov.au (J.B.); 7Anatomical Pathology, PathWest Laboratory Medicine, Royal Perth Hospital, Perth, WA 6000, Australia; 8Royal Victorian Eye & Ear Hospital, Melbourne, VIC 3000, Australia; daniel_w_mckay@hotmail.com; 9Ocular Oncology Unit, Sydney Eye Hospital and The Kinghorn Cancer Centre, Sydney, NSW 2000, Australia; robert.conway@sydney.edu.au; 10Save Sight Institute, The University of Sydney, Sydney, NSW 2000, Australia; 11School of Biomedical Sciences, University of Western Australia, Perth, WA 6000, Australia; bob.mirzai@uwa.edu.au (B.M.); wendy.erber@uwa.edu.au (W.N.E.)

**Keywords:** uveal melanoma, circulating tumour cells, liquid biopsy, CTCs

## Abstract

**Simple Summary:**

Approximately 50% of patients with uveal melanoma will develop incurable metastatic disease. This can be predicted, but requires a biopsy of the eye, which outside of centres of excellence, are not routinely performed. Given that uveal melanoma spreads through the blood, utilising circulating tumour cells from the blood might provide an alternative minimally invasive method to avoid the biopsy. However, the clinical application hinges on the detection rate of circulating tumour cells. Herein we assessed markers to improve the capture and detection of uveal melanoma circulating tumour cells and found that they could be detected in 86% of patients. We further found that ≥3 circulating tumour cells was significantly associated with worse survival.

**Abstract:**

(1) Background: The stratification of uveal melanoma (UM) patients into prognostic groups is critical for patient management and for directing patients towards clinical trials. Current classification is based on clinicopathological and molecular features of the tumour. Analysis of circulating tumour cells (CTCs) has been proposed as a tool to avoid invasive biopsy of the primary tumour. However, the clinical utility of such liquid biopsy depends on the detection rate of CTCs. (2) Methods: The expression of melanoma, melanocyte, and stem cell markers was tested in a primary tissue microarray (TMA) and UM cell lines. Markers found to be highly expressed in primary UM were used to either immunomagnetically isolate or immunostain UM CTCs prior to treatment of the primary lesion. (3) Results: TMA and cell lines had heterogeneous expression of common melanoma, melanocyte, and stem cell markers. A multi-marker panel of immunomagnetic beads enabled isolation of CTCs in 37/43 (86%) patients with UM. Detection of three or more CTCs using the multi-marker panel, but not MCSP alone, was a significant predictor of shorter progression free (*p* = 0.040) and overall (*p* = 0.022) survival. (4) Conclusions: The multi-marker immunomagnetic isolation protocol enabled the detection of CTCs in most primary UM patients. Overall, our results suggest that a multi-marker approach could be a powerful tool for CTC separation for non-invasive prognostication of UM.

## 1. Introduction

Uveal melanoma (UM) is the most common primary intraocular malignancy and the leading cause of death because of intraocular disease in adults [1]. Despite improvement in the treatment of the primary UM, overall survival has not changed significantly in the last 30 years [2]. Even with successful control of the primary tumour, incurable metastatic disease will ultimately develop in up to 50% of patients [3]. Extensive analysis of primary UMs has defined molecular features of the tumour cells that predict, with high accuracy, a patient’s risk for developing metastases. Highly predictive biomarkers of poor prognosis include histopathological features of the tumour, and chromosome alterations, including monosomy 3 together with 8q-gain [4,5], which correlate with the FDA approved gene expression profile test, DecisionDx-UM [6]. Although distinct biomarker profiles have been validated for personalised patient management, invasive surgical procedures with significant risk of sight-threatening complications are required in order to obtain sufficient tumour tissue for molecular analysis [7]. Routine implementation of less invasive strategies would enable early detection of metastasis and/or implementation of pre-emptive treatment strategies. 

Given that metastasis in UM arises from haematogenous dissemination, investigation of circulating tumour cells (CTCs) could provide a unique opportunity for genetic analysis of the patient’s tumour through a simple and safe blood test. CTCs have been previously detected in early-stage UM patients [8,9,10]. CTCs have been shown to constitute a source of tumour DNA that reflects the genetic landscape of the tumour of origin [10,11,12]. Therefore, CTCs can potentially detect tumour specific mutations and chromosomal copy number variations that predict the risk of metastasis for individual UM patients. To achieve this, CTCs need to be efficiently isolated in early-stage UM cases.

Current methods for isolating UM CTCs involve immunomagnetic capture and size-based filtration. However, the heterogeneous nature of UM cells may pose a significant barrier to the successful isolation and identification of CTCs from all patients, as the CTCs may express different markers [13]. Immunomagnetic capture of UM CTCs by targeting the melanoma-associated chondroitin sulphate proteoglycan (MCSP (also known as CSPG4, NG2)) protein was shown to successfully detect CTCs in patients with primary disease. Previous studies have shown detection of CTCs in 19% (1–5 CTCs per 50 mL of whole blood) [9], 14% (1–8 CTCs per 50 mL of whole blood) [8], and 69% (1–37 per 8 mL of whole blood) [10] of patients. Bidard et al. detected UM CTCs in only 30% of patients with metastatic disease using the CellSearch (Menarini Silicon Biosystems, Florence, Italy) system which targets the melanoma marker MCAM (melanoma cell adhesion molecule) and stains for MCSP [14]. A method using a dual marker enrichment protocol targeting CD63 (NKI/C3) and glycoprotein 100 (surface epitope), allowed for the detection of CTCs in 94% of patients with primary UM [15]. Thus, a multi-marker approach may be key to enriching the capture of CTCs from patients with UM. In fact, our previous study in metastatic cutaneous melanoma showed that targeting multiple membrane proteins resulted in the enrichment of a larger number of CTCs [16]. However, UM is clinically, phenotypically and genetically very different from cutaneous melanoma [17]. Thus, expression of potential markers for CTC isolation needs to be validated in UM tissue.

To enable greater efficacy and accuracy in capturing CTCs from UM patients, we systematically analysed the expression of several markers in a primary UM tumour microarray and in five UM cell lines. We then prospectively analysed a cohort of primary UM patients (n = 43) using a multi-marker capture assay and compared CTC detection rates to the results of our previously reported study (n = 26), in which MCSP alone was used for CTC capture.

## 2. Materials and Methods

### 2.1. Patients and Sample Collection

UM patients from the Lions Eye Institute, St John of God Hospital (Subiaco), and Royal Perth Hospital in Perth, Western Australia; The Royal Victorian Ear and Eye Institute, Melbourne, Victoria; and St Vincent’s Hospital, Sydney, New South Wales were enrolled in the study between March 2014 and July 2020. UM was diagnosed by clinical and ultrasound examination performed by a specialist ophthalmologist (ocular oncologists TI, RMC, DM) to evaluate the size and location of the intraocular tumour including the presence of ciliary body involvement. This study received approval from the Human Research Ethics Committee of Edith Cowan University (No. 11543 and No. 18957) and Sir Charles Gairdner Hospital (No. 2013-246 and No. RGS0000003289). Written consent was obtained from all patients under approved human research ethics committee protocols which complied with the Declaration of Helsinki. 

Peripheral blood samples were taken prior to radiation plaque insertion or enucleation. For CTC quantification, 8 mL blood was collected in either K_2_EDTA (BD, Franklin Lakes, NJ, USA) or TransFix CTC-TVTs (Cytomark, Buckingham, UK) tubes and processed within 1 h for EDTA, or 1–72 h for TransFix collected blood.

### 2.2. Immunohistochemistry

We performed a literature search and selected cell surface markers that could be used in combination to improve the number of CTCs captured and intracellular markers that could improve CTC identification (Appendix A).

Immunohistochemistry was performed on 4 µm sections cut from the formalin-fixed paraffin-embedded (FFPE) tissue microarray (TMA) block (Appendix A). Sections were deparaffinised in xylene followed by rehydration in graded ethanol for 3 min each then washed in running deionised H_2_O (dH_2_O) for 1 min. Antigen retrieval was performed in an 850W microwave oven for 15 min on 100% power in sodium citrate pH 6.0 buffer (gp100, MART1) or EDTA pH 8.0 buffer (MCAM, Nestin, ABCB5, RANK, 5HT2B, S100β, MCSP). Slides were then cooled for 8 min in running dH_2_O, permeabilised in Tris-Buffered Saline (TBS) containing 0.025% Triton X-100 (TX-100) in TBS (TBS: 50 mM Tris-Cl, 150 mM NaCl, pH 7.6) for 20 min then immunostained using an EnVision+ Dual Link System-HRP (DAB+) (Dako) according to the manufacturer’s instructions. Briefly, slides were incubated with Endogenous Enzyme Block for 10 min, rinsed 3 times for 5 min in TBS/0.025% TX-100 then incubated overnight at 4 °C with primary antibody diluted in TBS/1% Bovine Serum Albumin (BSA) (Appendix A). The following day, slides were washed 5 times for 5 min each in TBS/0.025% TX-100, incubated with Labelled Polymer-HRP for 30 min, rinsed 3 times for 5 min each in TBS/0.025% TX-100, incubated for 5 min with Substrate Chromogen, and then rinsed in dH_2_O. Slides were counterstained with weak Harris hematoxylin (Sigma) for 8 min, rinsed in running dH_2_O for 1 min, blued in 0.2% ammonia water for 2 min and mounted with ProLong Gold Antifade Mountant (Thermo Fisher Scientific). Immunostaining for each marker was evaluated by two independent (TV, ABB) unblinded observers as follows: negative (0), weak (1), moderate (2) or strong (3).

### 2.3. Cell Lines

The cell lines MM28, MP38, MP46, MP65 and MP41 exhibiting genetic profiles typical of UM were kindly donated by Prof Roman-Roman from the Institut Curie, France [18]. Cells were cultured in RPMI 1640 medium supplemented with 20% foetal bovine serum (FBS) at 37 °C in a humidified 5% CO_2_ incubator.

### 2.4. Flow Cytometry

MM28, MP38, MP41, MP46, and MP65 cells were harvested by incubation in 5 mM EDTA in RPMI 1640, resuspended then washed 3 times in Fluorescence-activated cell sorting (FACS) buffer (Phosphate-Buffered Saline (PBS) with 0.1% BSA, 25 mM HEPES, 1 mM EDTA, pH 7.0), incubated with primary antibody (Appendix A) for 30 min at 4 °C and washed 3 times in FACS buffer. Cells were then incubated with secondary antibody Alexa Fluor 488 conjugated donkey anti-rabbit or anti-mouse Ig (Abcam, Cambridge, UK) diluted 1:500 in FACS buffer for 15 min at room temperature (RT) and washed 3 times in FACS buffer prior to flow cytometric analysis on a Gallios Flow Cytometer (Beckman Coulter, Pasadena, CA, USA) and analysed with the Kaluza software package (Beckman Coulter). Comparison of the MCSP clones LHM2 and 9.2.27 was performed by flow cytometry, and no difference was observed in their binding to UM cells (Appendix A).

### 2.5. Immunocytochemistry

UM cell lines were fixed in PBS/4% paraformaldehyde for 10 min, washed 3 times for 5 min each in PBS, blocked and permeabilised in PBS/1% BSA/10% Normal Donkey Serum (NDS)/0.025% TX-100 for 20 min, then incubated overnight at 4 °C with primary antibody (Appendix A) diluted in PBS/1% BSA/1% NDS and washed 5 times for 5 min each in PBS/1% BSA/0.025% TX-100. Cells were then incubated for 1 h at RT with Alexa Fluor 488 conjugated donkey anti-rabbit or anti-mouse IgG (Abcam, Appendix A) diluted 1:500 in PBS/1% BSA/1% NDS, washed 3 times for 5 min in PBS/0.025% TX-100, mounted with ProLong Gold Antifade Mountant plus DAPI (ThermoFisher Scientific, Waltham, MA, USA) and analysed using an Olympus BX51 microscope equipped with an Olympus DP71 camera and DP Manager Software (Olympus, Shinjuku, Japan).

### 2.6. Circulating Tumour Cell Capture and Quantification 

Peripheral blood mononuclear cells (PBMCs) were isolated from 8 mL of blood by density gradient centrifugation over Ficoll-Paque (GE Healthcare, Chicago, IL) and resuspended in 1 mL Magnetic-Activated Cell Sorting (MACS) buffer (0.5% bovine serum albumin (BSA), 2 mM EDTA in PBS, pH 7.2) prior to the addition of 3 µL of each individual anti-ABCB5, gp100, MCAM, and MCSP coated immunomagnetic beads (Antibody coupling detailed in Appendix A). Cells and beads were incubated at 4 °C for one hour with rotation. Using a DynaMag-2 magnet (Life Technologies, Carlsbad, CA, USA), bead-captured cells were washed 3 times with MACS buffer, and then fixed with Medium A of the FIX & PERM Cell Permeabilization Kit (ThermoFisher Scientific) for 10 min at RT. Cells were washed twice in PBS, incubated in PBS containing 5% FcR Blocking Agent (Miltenyi Biotec, Bergisch Gladbach, Germany) for 10 min then incubated for 1 h at RT with anti-MART1/gp100/S100β and with anti-CD45 and CD16 antibodies conjugated with Alexa Fluor 647 diluted in Medium B of FIX & PERM Cell Permeabilization Kit (ThermoFisher Scientific)/2% NDS. After incubation, cells were washed in PBS, and incubated in 1:500 donkey anti-rabbit Alexa Fluor IgG 488 (ThermoFisher Scientific) diluted in Medium B/2% NDS/10 μg/mL Hoechst 33342 (Life Technologies) for 30 min at RT and placed on a magnet for 2 min. The resulting pellets were washed 3 times with PBS, resuspended in PBS, then mounted using Prolong Gold Anti-Fade reagent (Life Technologies). Slides were stored at 4 °C, visualised and scanned using an Eclipse Ti-E inverted fluorescent microscope (Nikon, Minato, Japan). Stained cells were analysed using the NIS-Elements Analysis software, version 4.2 (Nikon). CTCs were defined as nucleated cells (DAPI positive) that were positively stained for gp100/MART1/S100β, and negatively stained for CD45/CD16. 

### 2.7. Statistical Analysis

The Spearman rank correlation coefficient was used to test the correlation between the number of CTCs, tumour size, and age. The numbers of captured CTCs in patients with and without monosomy of chromosome 3 were compared using a non-parametric Mann-Whitney U-test. Statistical analyses were performed using R (The R Project, V 4.1.0).

Progression free survival (PFS) was defined as the time interval between blood collection and the date of first disease progression/death. Disease progression was determined by clinician assessment based on both radiological and clinical presentation of the patient. Overall survival (OS) was defined as the time between blood collection and death. OS and PFS were estimated using the Kaplan-Meier method and differences between curves were estimated using log-rank in the survival package in R (V 3.2-11) [19,20] and plotted using survminer (V 0.4.9) [21]. The cutp() function from survMisc (V 0.5.5) [22] was used to determine the optimal CTC cut-off of <3 and ≥3 for the Kaplan-Meier estimates.

## 3. Results

### 3.1. Analysis of Marker Expression in Primary UM Tumours

We assessed the expression of the selected markers (Appendix A) in a tumour microarray of 10 UM FFPE tumours. The markers used for CTC capture in our study were chosen based on earlier studies of their role in UM [9,10,15,23,24,25] and the evaluation of their expression in a UM TMA. ABCB5 is a known chemoresistance marker in cutaneous melanoma [26]; gp100 and MART1 are critical in melanosome biogenesis [27]; MCAM is an adhesion molecule, with some recent evidence suggesting that it plays a role in signalling [28]; MCSP can facilitate spreading of cutaneous melanoma, and lastly, S100β inhibits TP53 activity [29].

Detailed clinical characteristics of the tumour specimens included are described in Table 1. Antigen expression was scored according to the intensity of immunohistochemistry staining (Figure 1a,b). All tumour cores were assessable, except for tumour specimen PUM7, where only 3 of 4 cores could be scored. Representative images of positively stained cores for each of the markers are shown in Figure 1c. Duplicate cores were generally consistent in their staining intensity and the average intensity score per protein was calculated relative to the mean intensity score of all tumours that expressed the protein. For most cases, where strong staining for an individual marker was observed, that marker was homogeneously expressed in the tumour (Figure 1c). An exception was S100β where only a proportion of tumour cells exhibited strong expression, with positively stained cells often clustered in small areas (Figure 1c).

Overall, all tumours expressed gp100 and MART1, with an average expression intensity of 1.4 ± 0.4 and 2.5 ± 0.7 respectively (Table 2). Other intracellular markers, S100β and Nestin were expressed at similar intensity but in a lower proportion of tumours (70% and 60%, respectively). Interestingly, ABCB5 was strongly expressed in 90% of tumours (average expression intensity 2.7 ± 0.5). Of note, in addition to its well-characterised membranous location [30], ABCB5 was also localised within the cytoplasm of UM cells (Figure 1c). Other markers such as 5HT2B (1.1 ± 0.2), MCAM (1.6 ± 0.5) and RANK (1.3 ± 0.4) exhibited moderate levels of predominantly membrane-associated expression and were expressed in a lower proportion of tumour samples (Table 2). Interestingly, 5HT2B and MCAM expression coincided in various tumours (PUM5, 6, 10 and 9) and within areas of the tumours (Figure 1c). Strikingly, MCSP was not detected in any of the UM specimens but was strongly expressed in the cutaneous melanomas used as positive controls (Figure 1a).

### 3.2. Analysis of Marker Expression in Primary and Metastatic UM Cell Lines

We evaluated the expression of marker proteins in cell lines derived from primary (MP38, MP41, MP46, and MP65) and metastatic (MM28) UM. Flow cytometry was used to test the cell surface expression of 5HT2B, ABCB5, MCAM, MCSP and extracellular gp100 (referred to by its clone name, BETEB), to allow for accurate quantification of the percentage of cells expressing each marker. Immunocytochemistry was used for analysis of the intracellular markers, gp100, MART-1, Nestin, and S100β, as these markers could be used to confirm the identity of CTCs after immunocapture.

Flow cytometric analysis of UM cell lines revealed high levels of expression of cell surface MCSP, MCAM and gp100 (BETEB), with differential marker expression between cell lines (Figure 2). For example, MCSP, a biomarker commonly used to capture CTCs was expressed in all cell lines except for MP41. However, MP41 cell were strongly positive for MCAM and clearly positive for gp100 (BETEB). In contrast, 5HT2B and ABCB5 were expressed in only a small proportion of the cells within each cell line, apart from MP38, which exhibited low but homogeneous staining for 5HT2B (Figure 2).

We also assessed, by immunocytochemistry, the expression of intracellular markers which can be used for identification of CTCs and found that the melanocyte markers gp100 and MART1 were uniformly expressed in all cell lines. Nestin and S100β were expressed in a more heterogeneous pattern, with Nestin exhibiting high expression in MM28, MP38, and MP41 whilst MP46 and MP65 had medium levels of expression, and S100β only expressed in a subset of cells in each cell line (Figure 3).

### 3.3. Quantification of CTCs in Patients with Localised Disease Using Multi-Marker Immunomagnetic Beads

We next determined if the combination of ABCB5, gp100, MCAM, and MCSP would improve the rate of capture in samples from patients with UM. No suitable mouse monoclonal 5HT2B clone could be found and therefore this antibody was not included for CTC capture. We analysed the blood of 43 patients with localised primary UM (Table 3), without the presence of clinically identifiable metastatic disease. The blood sample was obtained before first-line therapy (radiotherapy or enucleation). CTCs were identified by positive staining of MART1/gp100/S100β (no suitable rabbit monoclonal Nestin clone could be found) and negative staining of CD45 and CD16. We found 37/43 (86%) patients with at least one detectable CTC in 8 mL of blood, with a range of 1–89 CTCs, and a mean and median of 7.5 and 3 CTCs, respectively. Representative images in Figure 4.

Tumour size, by apical height or basal diameter had no effect on the number of CTC detected detection of CTCs in patients (Figure 5a,b). The number of CTCs was comparable between anatomical location subgroups (Figure 5c). There was a negative correlation between age and CTC counts (Figure 5d). Amongst 19 patients with prognostic assessment by MLPA, the number of CTCs was higher and more variable in patients with high-risk UM tumours but was not significantly different (Figure 5e). Furthermore, we compared the number of CTCs recovered using the multi-marker assay to those previously obtained by targeting MCSP [10]. The detection rate was higher using the multi-marker assay (86%) compared to targeting MSCP alone (69%) (*p* = 0.14, Figure 5f).

### 3.4. Survival of Patients Stratified by CTC Count at Baseline

The median follow-up time of patients was 72 weeks (range 0–225) for the multi-marker cohort and 200 weeks (range 0–275) for the MCSP cohort (see Beasley et al., 2018 [10] for detailed cohort characteristics). Both cohorts were stratified into patients with <3 CTCs or ≥3 CTCs per 8 mL of blood. For MCSP captured CTCs, no significant association was found with OS (Figure 6a) or PFS (Figure 6b). On the other hand, CTCs captured using the multi-marker approach showed a significant association with OS (Figure 6c, *p* = 0.022) and PFS (Figure 6d, *p* = 0.04). The median OS for patients with ≥3 CTCs was 127 weeks, while no deaths were reported for those with <3 CTCs.

## 4. Discussion

Currently, outside of specialist ocular oncology centres, intraocular biopsies to determine prognosis are rarely performed. The analysis of CTCs may enable a minimally invasive alternative prognostication method, enabling closer follow-up of high-risk patients. Herein, we validated a panel of markers to capture CTCs in 86% of primary UM patients and demonstrated the association between CTCs and shorter PFS and OS.

Surprisingly, MCSP, which we and others have successfully used to capture CTCs, was not found to be expressed in any of the primary human UM tumour specimens included in our TMA. However, strong expression was observed in all but one of the UM cell-lines. Similar results of MCSP variable expression have also been reported on other UM cell lines [31]. MCSP is highly expressed in cutaneous melanomas, and although its expression is not well characterised in UM, a previous study has described its expression in approximately 95% (18/19) of primary UM tumours [32,33]. Reasons for the apparent lack of concordance between MCSP immunohistochemistry results and our successful use of MCSP antibodies to capture UM CTCs, along with results reported by Li et al. [32], are unknown. Notably, the MCSP antibody clone (9.2.27) used in our CTC capture protocol, and by Li et al. [32] the same, but differed from the antibody clone used here for immunohistochemical detection of MCSP (LHM2). Therefore, we evaluated the expression of MCSP in MP38 and MP41 cells by flow cytometry using the LHM2 clone and found results to be concordant with the 9.2.27 clone (Appendix A). Another explanation for our lack of apparent immunostaining of MCSP in our UM specimens may be because of the chemical fixation protocol, improper storage conditions, or length of storage [34,35]. The specific effects of these conditions on MCSP exposure are unknown. However, the other markers evaluated were positively stained in the same tumour samples.

The well-characterised melanoma antigens, MART1 and gp100 were expressed in all of our UM tumours, supporting previous studies of their detection in UM specimens and their potential inclusion in marker panels to capture UM CTCs [36]. Similarly, S100β was widely expressed but in a lesser proportion (70%) of our UM specimens, while MCAM, which was previously reported to be expressed in all of a cohort of 35 specimens [37], was expressed in just 4 of our 10 UM specimens. The TMA provides a useful means to evaluate biomarker expression. However, the relatively small areas of tumour tissue that can be evaluated in 1 mm cores may result in an underestimation of antigen expression if the immunohistochemical staining pattern is heterogeneous. For example, heterogeneous MCAM expression in UM described in a previous report [37] may have contributed to the lower proportion of MCAM-expressing tumours identified in the present study.

An interesting finding was that ABCB5 was expressed at elevated levels in a high proportion (90%) of UM specimens, predominantly localised to the cytoplasm. This contrasts with the sporadic expression of ABCB5 in cutaneous melanoma tumours [26,38]. ABCB5 is a cancer stem cell marker [39] over-expressed in cutaneous melanoma CTCs [38,40].

Biomarker expression was also examined in UM cell lines, which carry chromosomal losses and gains typical of human UM specimens [41,42]. Several of the markers, notably MCAM, MCSP and gp100 (BETEB) were expressed in most UM cell lines and in most cells in those cell lines. In contrast, 5HT2B and ABCB5, which were highly expressed in human UM tissue, were weakly and sporadically expressed in the UM cell lines analysed. Discordance of antigen expression between tumour tissue and cell lines could be attributed to environmental differences, and selection or adaptation to in vitro growing conditions [43].

More recently, microfluidics and sized based filtration have been used to isolate CTCs in various cancers [44,45,46], including cutaneous melanoma [47,48,49,50]. However, no studies have reported the isolation of UM CTCs using microfluidics. Mazzini et al. showed the isolation of UM CTCs using Isolation by Size of Epithelial Tumour cells (ISET). In that study, CTCs were detected in 17 of 31 (~58%) patients with localised disease [51]. This isolation rate was similar to detection of CTCs by MCSP alone in our previous study. Identification of CTCs following enrichment using a microfluidic device or other size-based separation method will still require immunostaining for antigens commonly expressed in UM. Thus, the results presented here could support the selection of markers to be used for UM CTC identification in future studies using any separation technology, including size-based separation or microfluidic devices.

Some of the earlier literature describing the associations between CTC enumeration and prognosis in UM are conflicting. Previous reports have identified that the numbers of CTCs found in patients with localised UM do not appear to correlate with prognosis or survival outcomes [8,15]. In contrast, other studies have found that CTC levels correlate with features of poor prognosis [9], reduced disease-free survival [51] and OS [52] in localised UM, or shorter PFS and OS in metastatic UM [14]. Here, we found that using our multi-marker approach, ≥3 CTCs were predictive of shorter OS and PFS. It should be noted that only a few patients progressed during the study time period, limiting the number of events. Given that the median follow-up time from diagnosis of the primary lesion is approximately 2.4–4.4 years [53,54,55,56] a longer-term follow-up study is required. However, we found here and in our previous study [10] that CTC numbers were not significantly associated with genome-based prognostic classes described previously [4]. One major limitation is that prognostic biopsies were only performed in a small subset of patients, restricting the assertiveness of this result.

While CTC enumeration appears to be promising for UM prognostication, our ultimate aim is to capture CTCs in most patients to provide sufficient cells for genetic analysis of the parental tumour. CTCs have been shown to constitute a source of tumour genetic material which represents that within the primary tumour [57]. Tura et al. [58] showed that chromosome 3 loss, a marker of poor prognosis, could be determined by fluorescence in-situ hybridisation in single CTCs and that these results matched the primary tumour in 10/11 cases. Furthermore, we have previously shown that CTCs from a patient with metastatic UM harboured virtually identical SCNAs as the primary excised tumour [10].

## 5. Conclusions

In summary, our study assessed the heterogeneity of expression of melanocytes, melanoma, and stem cell protein markers in UM tissues and cell-lines to select a panel of antibodies that could capture and identify CTCs in the blood of patients with primary UM. We found that when compared to our original study, where CTCs were captured using anti-MCSP alone, our multi-marker method improved both the number of patients with detectable CTCs, and the number of CTCs isolated. We further found that the enumeration of CTCs was significantly associated with shorter PFS and OS.

## Figures and Tables

**Figure 1 cancers-13-05990-f001:**
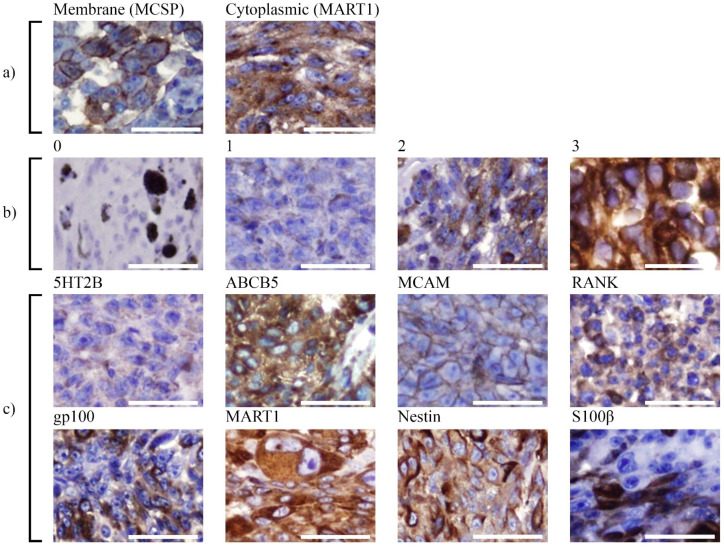
Immunohistochemistry of uveal melanoma. TMA staining demonstrating (**a**) Examples of how tumours were scored for protein localisation. Left shows strong membranous staining and right shows strong cytoplasmic staining. Representative image of positive MCSP membranous staining (left). (**b**) Examples of the criteria used to measure staining intensity, from 1 indicating weak staining to 3, the most intense staining. A tissue staining negatively by immunohistochemistry (0), is also shown (left), with intrinsic melanin pigment. (**c**) Shows a typical positive staining pattern for each of the markers analysed. Dark granular black spots are melanin deposits, and were not counted toward positive marker expression. All images taken at 200× magnification. Scale Bar = 50 μm.

**Figure 2 cancers-13-05990-f002:**
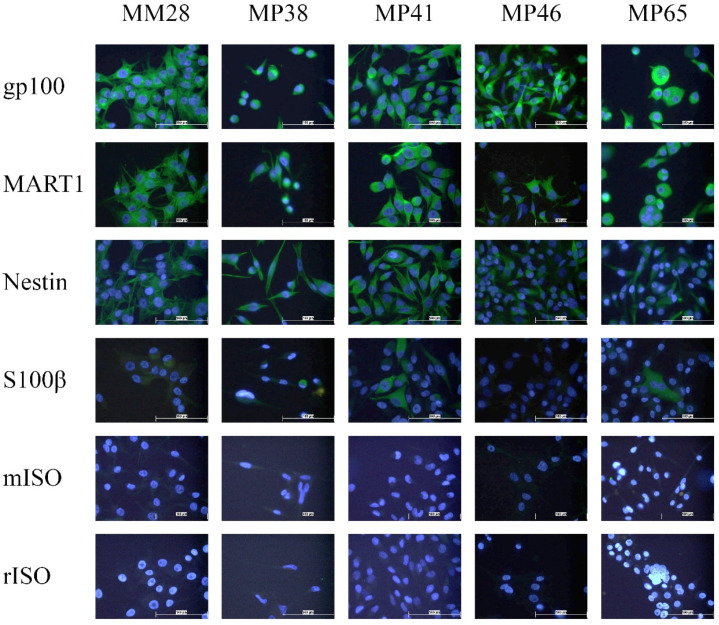
Immunocytochemical analysis of uveal melanoma cell lines. Intracellular markers and isotype controls. All images taken at 400× magnification. Scale bar denoting 100 µm. rISO—rabbit isotype control; mISO—mouse isotype control.

**Figure 3 cancers-13-05990-f003:**
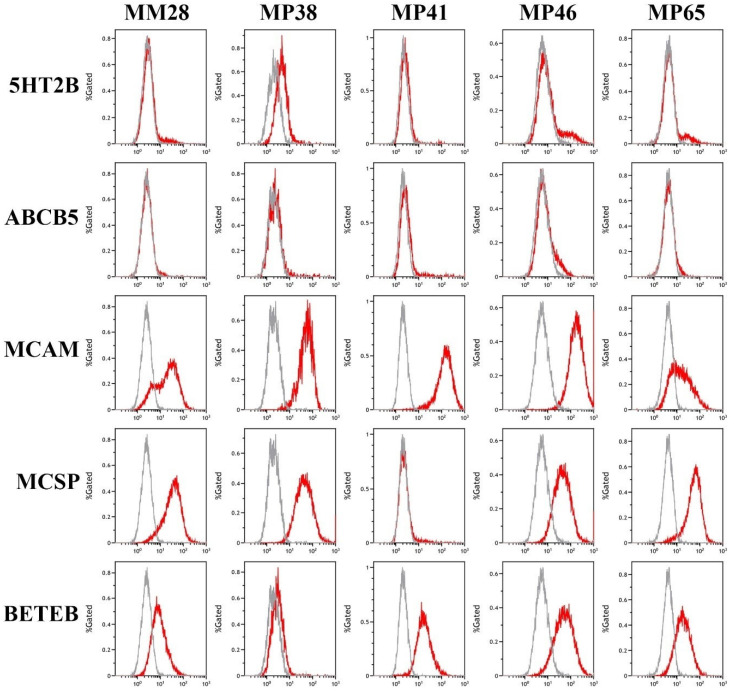
Expression of markers in cell lines by flow cytometry. Flow cytometric analysis of primary (MP38, MP41, MP46, and MP65) and metastatic (MM28) cell lines. Grey profiles represent negative controls using either rabbit or mouse IgGs depending on the primary antibody host.

**Figure 4 cancers-13-05990-f004:**
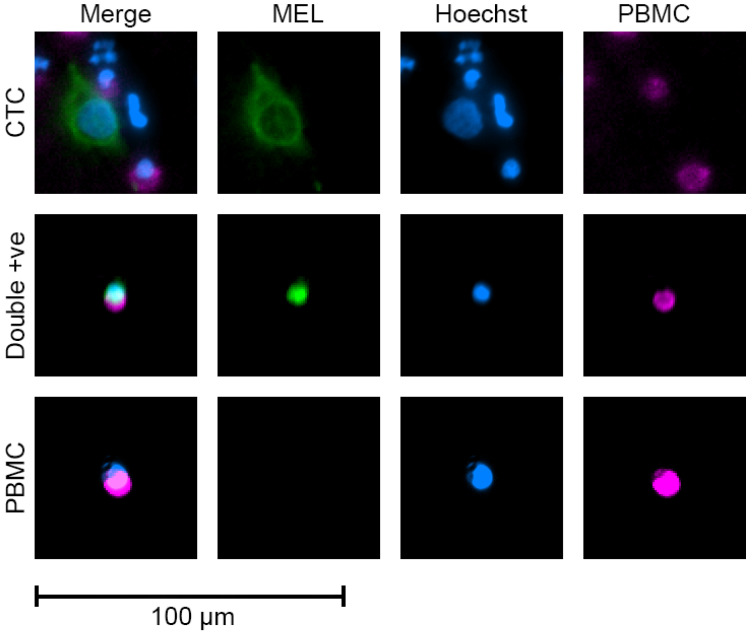
Example images of cells found in patient blood samples. Top to bottom: circulating tumour cell (CTC), double positive peripheral blood mononuclear cell (PBMC), and standard PBMC. CTCs were defined as melanoma (MEL: MART1, gp100, and S100β, green) and Hoechst (DNA, blue) positive, PBMC marker (CD45/CD16, purple) negative. Double positive PBMCs were not counted as CTCs, but were PBMCs (PBMC, purple) that strongly expressed melanoma (MEL, green) markers, and PBMCs were classified by the presence of CD45/CD16 (PBMC, purple).

**Figure 5 cancers-13-05990-f005:**
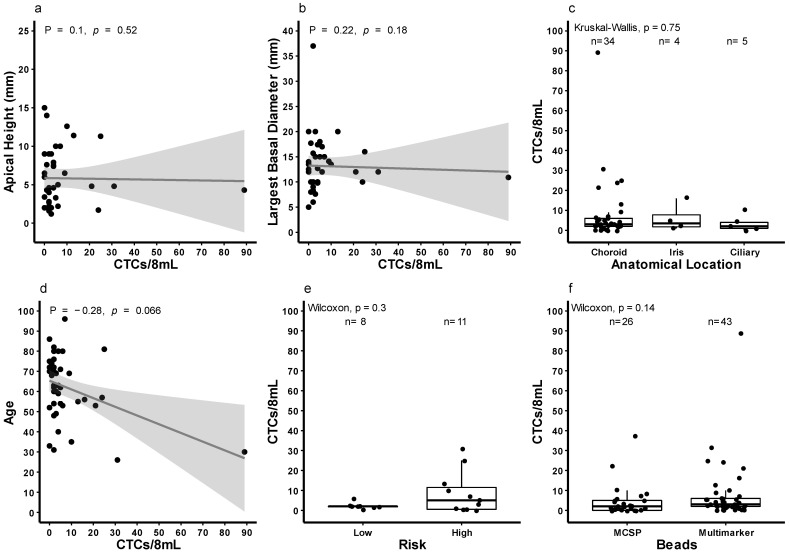
Correlation between circulating tumour cell numbers and patient and tumour features. Graphs illustrate Circulating Tumour Cell (CTC) counts between (**a**) apical tumour height (ρ = 0.1, *p* = 0.52), (**b**) largest basal diameter (ρ = 0.22, *p* = 0.18), (**c**) anatomical location (n = 43, *p* = 0.75), (**d**) age (ρ = −0.28, *p* = 0.066), (**e**) risk groups (n = 17, *p* = 0.3), and (**f**) the MCSP and multi-marker cohort (n = 69, Wilcoxon *p* = 0.14, χ^2^ *p* = 0.1694). Ρ in A, B, and D refers to Rho.

**Figure 6 cancers-13-05990-f006:**
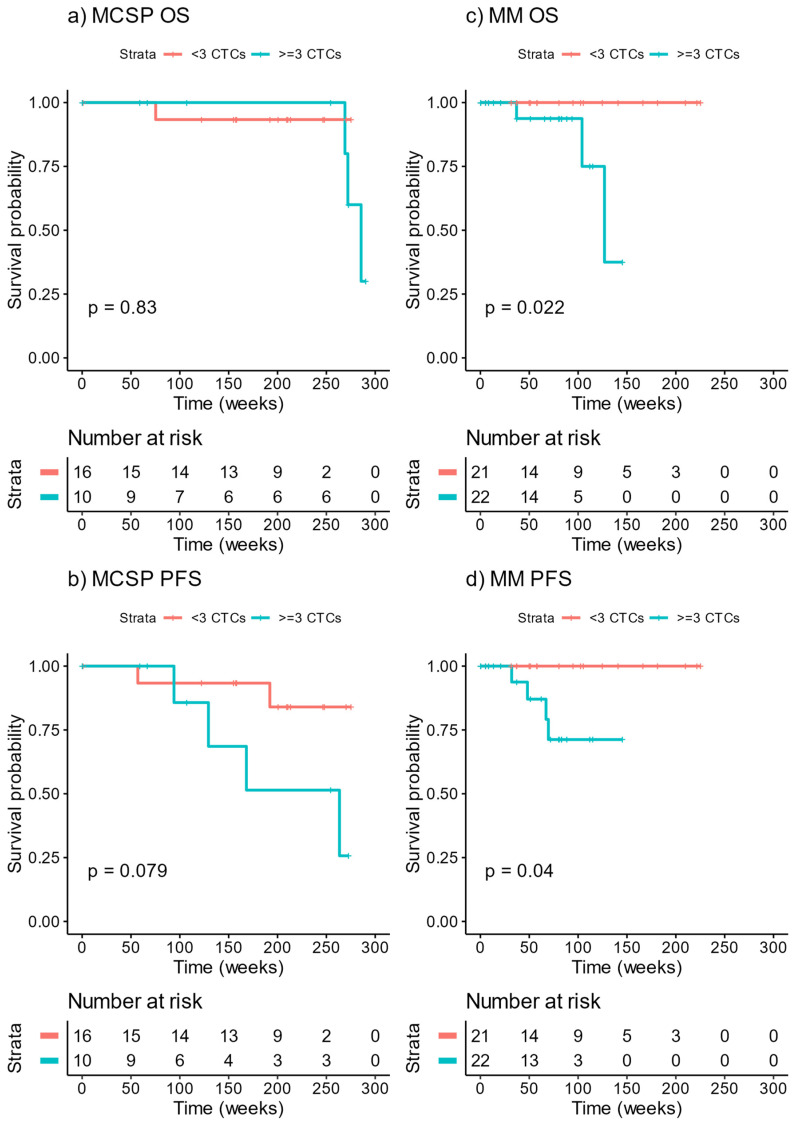
Overall and progression free survival of the MCSP and multi-marker cohorts. Kaplan-Meier estimates of (**a**) MCSP CTC OS, (**b**) MCSP CTC PFS, (**c**) multi-marker OS, and (**d**) multi-marker PFS grouped into <3 CTCs or ≥3 CTCs. OS—overall survival, PFS—progression free survival, MM—multi-marker. Log-rank *p* values are displayed on the graph.

**Table 1 cancers-13-05990-t001:** Clinical, Genotypic, and Histological Characteristics of Patient Tissue Samples Used on the Tissue Micro Array.

ID	Age *	Cell Morphology (Callender)	TMA Cell Morphology (Callender)	Tumour Size Basal × Height (mm)	Location	Metastatic Disease: Interval Time (Months)	MLPA
**PUM1**	38	M (Predominantly S)	M	12 × 9	Left choroidal	No: 51	NC **
**PUM2**	51	M	S/M	12 × 8	Right choroidal	Yes: 18	L3p, L3q, G8q
**PUM3**	65	M	E/M	10 × 6	Left choroidal	Yes: 33	L3p, G8q
**PUM4**	47	M	E/M	11 × 12	Right choroidal	Yes: −3	L3p, L3q, G8q
**PUM5**	59	M	S/M	9 × 4	Right choroidal	No: 42	G8q
**PUM6**	31	M (E < 10%)	E/M	15 × 12	Left choroidal	Yes: 12	G8q, some evidence of 3p loss, but not strong
**PUM7**	42	S	S	16 × 9	Left choroidal	Unknown	NC
**PUM8**	80	M	S	11 × 12	Left choroidal	Unknown	loss 1p, 3p, 3q, gain 8q
**PUM9**	59	M	E/M	14 × 8	Right choroidal	Yes: 3	loss 3p, 3q, gain 8q
**PUM10**	79	M	E	20 × 15	Right choroidal	No: 19	NC **

* Age at enucleation, ** Poorer quality DNA, M—mixed, S—Spindle, E—epithelioid, NC—no change, L—loss, G—gain.

**Table 2 cancers-13-05990-t002:** Average Marker Intensity Score.

Average Marker Intensity Score									
ID	ABCB5	MART1	gp100	S100β	Nestin	5HT2B	MCAM	RANK	MCSP
PUM5	3	3	1	2	2	1	1.5	0	0
PUM6	3	1	2	1.5	0	1	1	0	0
PUM10	2	3	1	2	2	1	2	2	0
PUM9	2	2	1	0	2	1.5	2	1	0
PUM2	3	2.8	1	2	1.5	1	0	0	0
PUM3	3	2.5	2	1	2	0	0	1	0
PUM4	2	2.6	1.3	0	0	1	0	1.3	0
PUM8	3	3	1.5	0	0	0	0	1	0
PUM7	3	3	1	1	0	0	0	0	0
PUM1	0	2	2	1	0	0	0	0	0

0—no, 1—low, 2—moderate or 3—high antigen expression.

**Table 3 cancers-13-05990-t003:** Clinical characteristics of the multi-marker CTC cohort.

PID	Age	Sex	Eye	Anatomical Location	Apical Height (mm)	Largest Basal Diameter (mm)	Volume (mm^3^)	Callendar Classification	BAP1 Status	Tissue Mutation	Genetic Features	CTCs	FU (Weeks)	PFS (Weeks)	COD
556	59	F	Left	Choroid	4.6	9.8	462	Mixed	-	-	-	4	72		
565	72	M	Left	Choroid	4	8.3	288	-	-	-	NC	2	221		
580	80	F	Left	Choroid	4.6	10	481					2	225		
646	33	F	Right	Choroid	2	5	52					0	210		
694	68	M	Left	Choroid	4.3	10	450					1	141		
695	80	F	Left	Choroid	7.9	17.4	2503					4	37	32	UM
703	96	F	Left	Choroid	10	15	2355	Mixed	L	*GNAQ Q209P*	L1p, L3, L6q, G8	7	66		
712	57	F	Right	Choroid	1.7	10	178					24	13		
716	72	F	Left	Iris	-	-	-	Spindle	-	*GNAQ Q209L*	NC	1	166		
721	49	F	Left	Choroid	2	15	471	Mixed	P	*GNA11 Q209L*	L1p, L3, L8p, G8q	3	127	70	UM
763	63	M	Right	Ciliary	2.8	6	106					2	105		
805	82	M	Left	Choroid	2	9	170	Mixed				2	181		
840	81	M	Right	Choroid	11.3	16	3028	Mixed	L	-	L3, G8q	25	104	67	UM
872	74	M	Right	Choroid	7.6	17.7	2492	Mixed	-	*GNA11 Q209L*	-	1	125		
879	35	F	Right	Ciliary	12.6	13.5	2404	Epithelioid	-	*GNAQ Q209P*	pL3, G8q	10	145		
995	76	M	Left	Choroid	9	10	942		-	*GNAQ Q209L*	-	2	0		
1022	71	F	Right	Iris	-	-	-					5	83		
1041	26	M	Right	Choroid	4.8	12	723	Mixed	L	*GNAQ R183Q*	L3, L6q, L8p, G8q	31	115		
1059	69	M	Right	Choroid	9	20	3768	Mixed	-	*GNAQ Q209P*	-	3	112		
1160	62	F	Left	Iris	1.6	37	2293	Mixed	P		pG6p	2	58		
1163	48	M	Left	Choroid	2.2	10	230	-	-	-	NC	2	102		
1180	53	M	Left	Choroid	5	17	1512	Mixed	P	*GNAQ Q209P*	NC	6	83		
1217	70	F	Right	Choroid	4.6	15.7	1187					2	51		
1218	63	F	Left	Choroid	7.4	12.7	1249	Spindle	P	-	-	4	51		
1227	40	F	Left	Ciliary	7.5	10	785					4	88		
1231	31	F	Right	Choroid	2.7	9	229					2	95		
1237	62	F	Right	Choroid	3.3	15	777	-	-	*GNAQ Q209P*	L1p, pL*BAP1*, pL6p	5	80		
1238	80	M	Right	Choroid	2.2	12	332					6	21		
1258	54	M	Right	Choroid	10	18	3391	Mixed	P	*GNAQ Q209L*	G6p, G8q	5	81		
1284	54	F	Right	Choroid	2	10	209	Mixed	-	-	L1p, G6p	2	31		
1285	70	F	Left	Choroid	9	14	1846	Mixed	L	*GNAQ R183Q*	L1p, L3, G8q	0	58		
1286	52	M	Right	Choroid	3.4	12.5	556	Epithelioid	L	-	-	0	37		
1287	30	M	Right	Choroid	4.3	10.9	535	Epithelioid	P	-	-	89	37		
1297	69	F	Left	Choroid	6.5	14.1	1353					9	8		
1298	75	M	Left	Choroid	6	12	904	-	-	*GNA11 Q209L*	G6p	0	80		
1307	56	M	Left	Iris	-	-	-					16	94	48	
1308	73	M	Right	Ciliary	14	8	938	-	-	*GNAQ Q209L*	L1p, L3, G8	1	50		
1338	72	M	Right	Choroid	6.5	13.5	1240	-	-	-	L1p, L3, G8	0	5		
1370	60	M	Left	Choroid	2.7	10	283	Spindle	P	*GNAQ Q209P*	G6p	2	0		
1387	55	M	Right	Choroid	11.4	20	4773	Mixed	L	-	L3, G6p, L6q, G8q	13	5		
1401	53	F	Right	Choroid	4.8	12	723					21	0		
1405	60	F	Right	Choroid	1.2	7.6	73					3	0		
1408	86	M	Left	Ciliary	15	20	6280	Mixed	-	*GNAQ Q209L*	L1p, L3, pL6p, G8q	0	37		

F—female; M—male; L—loss; P—present; NC—no change; FU—follow-up time; p—partial; G—gain; L—loss. UM—uveal melanoma; -—unknown. COD—cause of death.

## Data Availability

Data sharing not applicable. All tabulated data is available within the manuscript.

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
