# Peer review of "Analysis of Circulating Tumour Cells in Early-Stage Uveal Melanoma: Evaluation of Tumour Marker Expression to Increase Capture"

_cancers, 2021, doi:10.3390/cancers13235990_

Round 1

Reviewer 1 Report

None

This is an outstanding report describing the expression of melanocytes, melanoma, and stem cell protein markers in uveal melanomas (UM) tissues and cell-lines. The investigators identified  a panel of  antibodies that could capture and identify circulating tumor cells (CTCs) in the blood of patients with primary UM. The multi-marker method improved both the number of patients with detectable CTCs, and the number of CTCs isolated. They also further found that the enumeration of CTCs was significantly associated with shorter Progression free survival and overall survival. The studies are carefully done with complete reference to other studies in this field.

Author Response

We thank reviewer #1 for their review and kind comment.

Reviewer 2 Report

The heterogeneity expression of several protein markers from melanocytes, melanoma and stem cell was evaluated in UM tissues and cell-lines in this paper. Based on the analysis, the authors chose a panel of antibodies and successfully captured and identified CTCs from the blood of patients with primary UM. Notably, the detection rate by multimarker assay was significantly increased from 69% to 86% when compared to the classical assay by targeting MSCP. After throughout evaluation of the research article, I personally felt that the presented article is good, and it fits with the scope of the Cancers due to the research work aims to address an important topic for researchers, clinicians, and patients. Some comments and suggestions are presented below to improve quality and to clarify some information.

  1. Figure 2 and Figure 3 are messed up: the image of current figure 3 should be exchanged with the image of current figure 2. “mISO” and “rISO” were not mentioned in anywhere of the paper.
  2. PID 1287 patient has 89 CTCs been detected. This number of detected CTCs is dramatic higher than all the other patients’ result. I am worried about the data of PID 1287 patient will significantly affect all the correlation analysis in figure 5. Can the authors try to explain on this concern or perform some analysis to minimize this potential issue?
  3. The multimarker assay is more sensitive than the MSCP-targeting assay. That means with the same detected number of CTCs, that should be more CTCs in the blood of patients who treated with MSCP-targeting assay. These patients are supposed to have a shorter OS. However, in figure 6, it looks like MM OS is shorter than MSCP OS. Please try to clarify on this concern.
  4. A brief introduction on key markers used for multimarker will be nice.

Author Response

Reviewer #2

We thank reviewer #2 for their review and comments. Please see responses to queries:

Changes in text have been highlighted in red.

The heterogeneity expression of several protein markers from melanocytes, melanoma and stem cell was evaluated in UM tissues and cell-lines in this paper. Based on the analysis, the authors chose a panel of antibodies and successfully captured and identified CTCs from the blood of patients with primary UM. Notably, the detection rate by multimarker assay was significantly increased from 69% to 86% when compared to the classical assay by targeting MSCP. After throughout evaluation of the research article, I personally felt that the presented article is good, and it fits with the scope of the Cancers due to the research work aims to address an important topic for researchers, clinicians, and patients. Some comments and suggestions are presented below to improve quality and to clarify some information.

Figure 2 and Figure 3 are messed up: the image of current figure 3 should be exchanged with the image of current figure 2. “mISO” and “rISO” were not mentioned in anywhere of the paper.

We thank reviewer #2 for noticing this mistake. This has been corrected and in figure legend 2 we have added the following text “rISO – rabbit isotype control; mISO – mouse isotype control.”

PID 1287 patient has 89 CTCs been detected. This number of detected CTCs is dramatic higher than all the other patients’ result. I am worried about the data of PID 1287 patient will significantly affect all the correlation analysis in figure 5. Can the authors try to explain on this concern or perform some analysis to minimize this potential issue?

The single patient does not greatly affect the correlations found in figure 5. Removal of the outlier does not affect the outcome of the analysis (i.e. not significant). Given that this is true biological data, we do not think that it is appropriate to remove outliers for analysis.

The multimarker assay is more sensitive than the MSCP-targeting assay. That means with the same detected number of CTCs, that should be more CTCs in the blood of patients who treated with MSCP-targeting assay. These patients are supposed to have a shorter OS. However, in figure 6, it looks like MM OS is shorter than MSCP OS. Please try to clarify on this concern.

The patients in the MCSP cohort, as stated in the manuscript as Beasley et al., 2018 were not the same as the patients in the multi-marker cohort. In essence, we show long term follow up for our MCSP cohort showing that using that method CTCs were not prognostic (although they do seem to be associated with more events, there was no statistical difference between the groups) whereas CTCs isolated with the multimarker cohort are significantly associated with prognosis. While we noted these differences, we cannot directly statistically compare these two cohorts.

A brief introduction on key markers used for multimarker will be nice.

We have added a very minor introduction to the markers used.

Page 6, lines 217-224: ABCB5 is a known chemoresistance marker in cutaneous melanoma [27]; gp100 and MART1 are critical in melanosome biogenesis [28]; MCAM is an adhesion molecule, with some recent evidence suggesting that it plays a role in signalling [29]; MCSP can facilitate spreading of cutaneous melanoma, and lastly, S100β inhibits TP53 activity [30].

With the following references added:

  1. Frank, N.Y.; Margaryan, A.; Huang, Y.; Schatton, T.; Waaga-Gasser, A.M.; Gasser, M.; Sayegh, M.H.; Sadee, W.; Frank, M.H. ABCB5-mediated doxorubicin transport and chemoresistance in human malignant melanoma. Cancer Res 2005, 65, 4320-4333, doi:10.1158/0008-5472.can-04-3327.
  2. Hoashi, T.; Watabe, H.; Muller, J.; Yamaguchi, Y.; Vieira, W.D.; Hearing, V.J. MART-1 is required for the function of the melanosomal matrix protein PMEL17/GP100 and the maturation of melanosomes. J Biol Chem 2005, 280, 14006-14016, doi:10.1074/jbc.M413692200.
  3. Wang, Z.; Xu, Q.; Zhang, N.; Du, X.; Xu, G.; Yan, X. CD146, from a melanoma cell adhesion molecule to a signaling receptor. Signal Transduct Target Ther 2020, 5, 148, doi:10.1038/s41392-020-00259-8.
  4. Eisenstein, A.; Gonzalez, E.C.; Raghunathan, R.; Xu, X.; Wu, M.; McLean, E.O.; McGee, J.; Ryu, B.; Alani, R.M. Emerging Biomarkers in Cutaneous Melanoma. Mol Diagn Ther 2018, 22, 203-218, doi:10.1007/s40291-018-0318-z.